# In-Situ Monitoring and Analysis of the Pitting Corrosion of Carbon Steel by Acoustic Emission

**Junlei Tang** [1] **, Junyang Li** [2] **, Hu Wang** [2,*] **, Yingying Wang** [1] **and Geng Chen** [3]

[1] School of Chemistry and Chemical Engineering, Southwest Petroleum University, Chengdu 610500, China; tangjunlei@126.com (J.T.); yingyingwanglyon@126.com (Y.W.)
[2] School of Material Science and Engineering, Southwest Petroleum University, Chengdu 610500, China; lijunyang0607@126.com
[3] CNOOC Energy Technology & Services-Shanghai Environmental Engineering & Technology Branch, Shanghai 200335, China; chengeng@cnooc.com.cn
[*] Correspondence: senty78@126.com; Tel.: +86-028-83037361

**Abstract:** The acoustic emission (AE) technique was applied to monitor the pitting corrosion of carbon steel in $NaHCO_3$ + $NaCl$ solutions. The open circuit potential (OCP) measurement and corrosion morphology in-situ capturing using an optical microscope were conducted during AE monitoring. The corrosion micromorphology was characterized with a scanning electron microscope (SEM). The propagation behavior and AE features of natural pitting on carbon steel were investigated. After completion of the signal processing, including pre-treatment, shape preserving interpolation, and denoising, for raw AE waveforms, three types of AE signals were classified in the correlation diagrams of the new waveform parameters. Finally, a 2D pattern recognition method was established to calculate the similarity of different continuous AE graphics, which is quite effective to distinguish the localized corrosion from uniform corrosion.

**Keywords:** carbon steel; pitting corrosion; acoustic emission; wavelets; pattern recognition

---

## 1. Introduction

In the modern construction industry, carbon steel is widely used because of its low price and good mechanical properties. However, its corrosion resistance in an aqueous environment is very low. Especially in the presence of a saline medium, the probability of corrosion increases greatly, mostly including localized corrosion. Structural damage and mechanical performance degradation can easily happen during long term service with localized corrosion. As one of the most destructive forms of localized corrosion, pitting corrosion of carbon steel intensively occurs in many sites, such as steel in reinforced concrete, and steel with degraded coatings of a bridge or water containment structure. Many investigations [1–3] have been done for the detection of structural failure of concrete with the acoustic emission (AE) technique. As the initiation stage of many structural degradation cases, the pitting corrosion of carbon steel needs to be monitored.

It is commonly believed that pitting corrosion happens on passivated metals and alloys in a solution of halide ions [4]. The properties of the passive film plays an important role in the initiation, growth, and re-passivation of pits. In most occasions, the pitting process is regarded as containing several stages: (1) Local breakdown of the passive film, represented as the nucleation process, (2) propagation (accelerated corrosion), (3) stable growth, and (possibly) (4) re-passivation. Stainless steels are typically passive initially. The initiation of pitting is caused by the adsorption of halide ions through the passive film. Additionally, the growth of the pit occurs mainly at the bottom as the anodic dissolution inside the pit. The cathodic reaction is nevertheless outside the pit,

on inclusions or other defects. Stainless steels are relatively highly resistant to pit initiation, which can be ascribed to the excellent passivity of the $\gamma$-Fe$_2$O$_3$ plus Cr$_2$O$_3$ oxide film at the surface. Hence, only a limited amount of pits can be formed on the stainless steel surface. However, when the pit forms, it often grows rapidly and develops in depth. The aggregation of corrosion products at the pit mouth accelerates the corrosion attack beneath the occluded area. The rupture or breakdown of the cover at the pit mouth, including the corrosion products and passive film, may result in re-passivation, which implies the end of the pitting corrosion. As for other materials with a less pronounced passive film, such as carbon steels, pitting corrosion may start at inconsistent pores of the passive film or corrosion products [5]. Different from stainless steels, the growth of pitting undergoes at a relatively slow rate. Pits may repeatedly experience the initiation—growth—re-passivation process [6]. In many occasions, the corrosion attack consists of many shallow pits in the nearby area. The re-passivation of the pit is most probably caused by the passivator in the solution, not by the metal itself. The rupture or breakdown of the passive film and product film is not as notable as stainless steel.

It is generally assumed that the iron oxide and ferrous bicarbonate can form as the passive film of carbon steel in a carbonate solution [5,6]. The origin of the passive film is from the oxidation of iron by dissolved oxygen and the deposition of insoluble ferrous bicarbonate. Many studies have focused on the structure and composition of the passive film. Although there are two models of the passive film (the $\gamma$-Fe$_2$O$_3$ layer and the $\gamma$-Fe$_2$O$_3$ plus Fe$_3$O$_4$ double layer), it is not surprising that the passive film on iron mainly consists of $\gamma$-Fe$_2$O$_3$. It is also more acceptable that the passive film of carbon steel in a bicarbonate solution consists of an inner Fe$_3$O$_4$ layer and an outer $\gamma$-Fe$_2$O$_3$ layer, both combined with insoluble ferrous bicarbonate [5]. The inhomogeneity of the passive film on carbon steel is the main reason for the initiation of pitting corrosion.

Many techniques have been used to study pitting corrosion. The American Society for Testing and Materials (ASTM) has standardized methods for the study/evaluation of pitting corrosion and pitting tests in 6% FeCl$_3$. Electrochemical measurements are most commonly used in investigating pitting corrosion behavior, e.g., polarization curves measuring the break potential of pitting, E$_b$; protective potential of pitting, E$_p$; and the free corrosion potential, E$_{corr}$. Current fluctuation behavior in potentiodynamic or potentiostatic polarization is extensively investigated to probe metastable pitting, which is widely regarded as an important phenomenon before stable pitting and can be used to predict stable pitting events. Electrochemical impedance spectroscopy (EIS) is also widely used in the evaluation of the properties of the passive film under pitting attack. Moreover, electrochemical noise (EN) is another useful technique in the investigation of pitting events and the monitoring of pitting corrosion. Although many approaches have been utilized in corrosion investigation, only limited techniques, like EN, linear polarization resistance probe [7,8], electrical resistance probe [9], and the field signature technique [10], can be applied in pitting monitoring. All of them, however, have their limitations. For example, EN is quite sensitive in in-situ recording and identifying the initiation of pitting events [11,12]. However, it is very difficult to interpret the data of the propagation process. Therefore, in-situ monitoring and the prediction of pitting corrosion events in service is still a great challenge in industrial applications.

Acoustic emission (AE), an important technique that can be used in corrosion monitoring, has attracted great attention in recent years. In most occasions, the corrosion of metals and alloys is always accompanied by a rapid release of energy in the form of a transient elastic wave. The nature of corrosion monitoring by AE is firstly composed of the relationship between the corrosion attack and the transient elastic wave. Such a relationship is then used for in-situ monitoring of the corrosion attack of the material in service. AE is a non-destructive technique, which aims at in-situ monitoring of corrosion attack by detecting, recording, and analyzing the acoustic emission signal. The modern AE technique began with the work of Kaiser in Germany in the 1950s. In recent years, AE has been widely used in many applications, such as the petroleum and natural gas industry, aerospace industry, transportation industry, and construction industry, etc. [13–16].

Many types of corrosion have been studied by AE [17–25], such as stress corrosion cracking [18–20], abrasion or erosion corrosion [21], and pitting corrosion [22–25]. Mazille [22] proved that AE signals can be easily detected in pitting corrosion. Moreover, a good correlation between AE activity and the pitting rate was observed. Then, the initiation and propagation of pitting corrosion on stainless steels was studied with AE [23]. It further demonstrated that the initiation step of pitting corrosion was not significantly emissive, whereas the propagation step was characterized by the emission of resonant signals. They ascribed the signals in the propagation step to the evolution of hydrogen bubbles. More recent studies by other researchers, such as Darowicki, used AE and potentiodynamic methods to investigate the mechanism of pitting corrosion of stainless steel [24]. Wu and Jung used AE to monitor the pitting corrosion on vertically positioned 304 stainless steel, and analyzed the acoustic emission energy parameter [25]. However, most researchers used some methods, such as potentiodynamic polarization, potentiostatic polarization, or heating, to accelerate the pitting corrosion of metals in a short time, thus not realistically simulating the corrosion process of materials under natural conditions. It is very important that the corrosion behavior of steel through in-situ monitoring of the pitting corrosion process under natural conditions is understood to implement structural integrity management for steel construction.

The processing and analyzing of the AE signal is the most important step in AE studies. In many studies, the acoustic emission signals are processed and classified by various methods, like K-means, random forest, wavelet analysis, or some other parameter analysis methods [26–33]. The application of some methods, for instance, the K-means and random forest, only utilize some parameters of the AE waveform (raw or pre-treated). The single use of a wavelet can only optimize AE waveforms. Therefore, there remains much work to do for the development of an effective and integrative method to identify the corrosion type based on raw data from in-situ AE monitoring.

In this paper, the pitting corrosion of carbon steel was monitored by the AE technique in $NaHCO_3$ + NaCl solutions, in which the pitting initiation—growth—re-passivation process was present [5]. Open circuit potential experiments were carried out simultaneously to probe the relationship between the AE and electrochemical behavior. In addition, an optical microscope was used to obtain in-situ recordings of the surface morphology of carbon steel under corrosion attack in different conditions and SEM was used to observe the surface micromorphology after corrosion. Finally, Matlab was involved to establish the methods for AE data processing [33] and analyzing.

## 2. Materials and Methods

### 2.1. Material Preparation

Q235 carbon steel was used for the measurements and the chemical composition of the specimens is listed in Table 1. The specimens were cut out from a cold rolled sheet of 2 mm thick steel before measurements and coated with a high-temperature resistant silicone gasket to keep the exposed surface as 15 mm × 15 mm. The specimens were abraded gradually from 180 to 1000 grit silicon carbide paper. Then, they were rinsed with deionized water and acetone, and were dried and stored before use.

**Table 1.** Composition of Q235 carbon steel.

| C | Mn | Si | S | P |
|---|---|---|---|---|
| ≤0.22% | 0.3%–0.65% | ≤0.35% | ≤0.05% | ≤0.045% |

### 2.2. Corrosion Conditions

The corrosion solution consisted of 2000 mg/L $NaHCO_3$ in the presence of different concentrations of NaCl, ranging from 500 mg/L to 1200 mg/L. The pH value and temperature used in the corrosion experiments was 6.7 and 25 °C, respectively.

### 2.3. Setup of Acoustic Emission Monitoring

The AE acquisition system and pitting assembly are shown in Figure 1. The AE acquisition system consisted of a Mistras USB AE Node, sensor R15α (50–400 kHz). The sensor was assembled by spring applying 6 N of force. The high vacuum grease was used between the interface of the sensor and the specimen to ensure the best connection. The AE acquisition options are shown in Table 2.

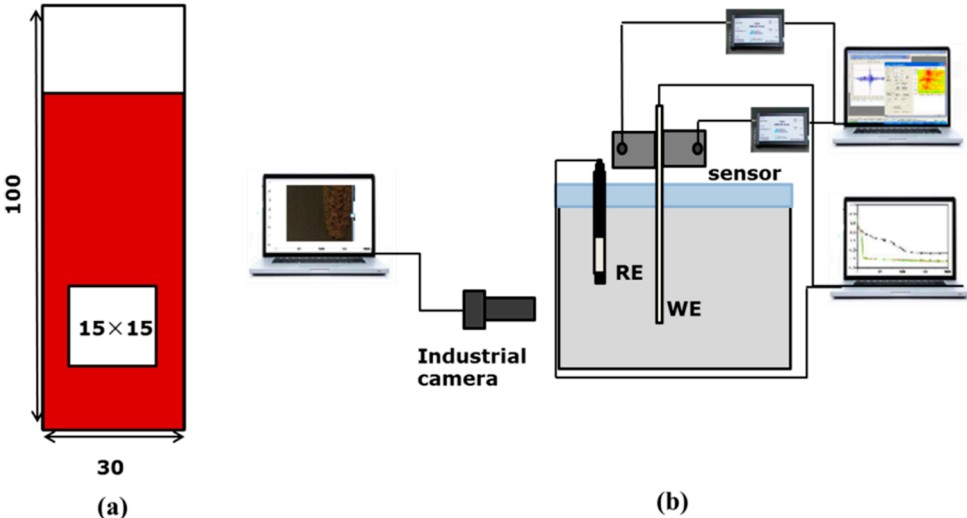

**Figure 1.** Schematic diagram of the acoustic emission (AE) monitoring of pitting corrosion: (**a**) Q235 specimen (dimension in mm), (**b**) pitting assembly, and the AE system and potential measurement.

**Table 2.** AE acquisition options.

| Threshold | PDT | HDT | HLT | Analog Filter | Sample Rate | Pre-Trigger | Length |
|---|---|---|---|---|---|---|---|
| 27 dB | 200 μs | 400 μs | 200 μs | 100 K–400 KHz | 2 MPS | 40 μs | 2 K |

### 2.4. Electrochemical Measurements and Corrosion Morphology Observations

Open circuit potential (OCP) was measured by a CHI4 electrochemical workstation of Wuhan Corrtest Instruments Corp. LTD in China. Specimens of Q235 carbon steel were used as the work electrodes. A saturated calomel electrode (SCE) was used as the reference electrode. The corrosion morphologies were recorded by an industrial digital camera (Yilong Electronic Technology Co., Ltd., Shenzhen, China, 5 mega-pixel) during the experiment and a scanning electron microscope (SEM, ZEISS, EVO, MA 15) after the experiments. The OCP and AE measurements were carried out and recorded simultaneously.

## 3. Results

### 3.1. AE and OCP Monitoring

Figure 2 shows the OCP and AE monitoring of the corrosion process of Q235 carbon steel in 2000 mg/L $NaHCO_3$ in the presence of different concentrations of NaCl. In Figure 2a, OCP behaved differently with time in various NaCl concentrations. The OCP shifted slightly towards the positive direction and gradually stabilized after 50,000 s of immersion in the presence of 500 mg/L NaCl, indicating that the interface condition was influenced by the presence of NaCl. With the increase of the NaCl concentration, the OCP behavior changed. In 800 mg/L NaCl, the OCP shifted negatively and could not reach a stable state in approximately 90,000 s immersion time. At the NaCl concentration of 1000 mg/L NaCl and 1200 mg/L, the OCP moved sharply towards the negative direction and reached a relative stable state at 65,000 s and 30,000 s, respectively. The above results revealed the different

interface conditions of carbon steel in NaHCO$_3$ solution in the presence of different concentrations of NaCl. It was commonly believed that the iron oxide and ferrous bicarbonate form as the passive film of carbon steel in bicarbonate solutions [5,6]. The addition of chloride iron accelerated the breakdown of the passive film and triggered the initiation of pitting. Thus, the higher the concentration of Cl$^−$, the faster the passive film is broken and the OCP is stabilized.

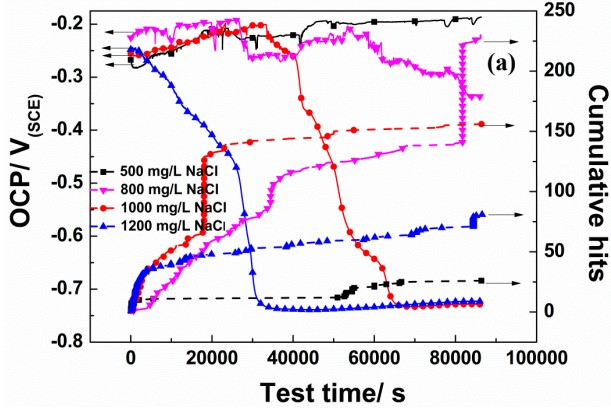

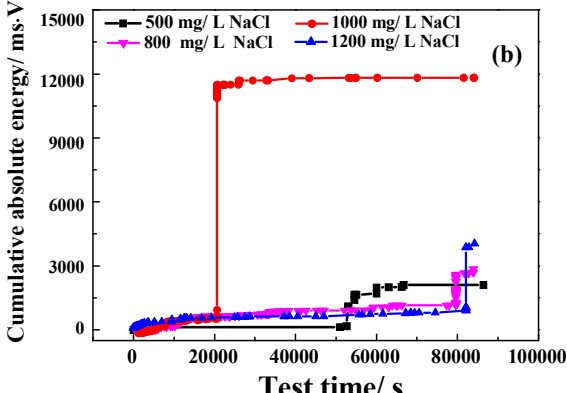

**Figure 2.** The open circuit potential (OCP) and AE monitoring of the pitting corrosion of carbon steel in NaCl + NaHCO$_3$ solutions: (**a**) OCP and cumulative hits versus time, (**b**) cumulative absolute energy versus time.

Figure 2a also presents the influence of the NaCl concentration on cumulative hits of acoustic emission and the correlation between the OCP and cumulative hits. In the concentration of 500 mg/L, the cumulative hits remained at very low level during the first 5000 s and then started to increase slightly, indicating the corrosion events occurring such Cl$^−$ concentrations were not very noticeable. The few cumulative hits in the AE monitoring is in accordance with the relatively small variation range of the OCP.

The cumulative hits continuously increased with time in the presence of 2000 mg/L NaHCO$_3$ + 800 mg/L NaCl, which represented the occurrence of corrosion events all the time. Repeating metal dissolution beneath the cover of the pits (occluded cell) and the initiation of new pits resulted in the continuous [6] increasing of cumulative hits. Most likely, the breakdown of the corrosion product film upon the occlude cell induced the sudden ascent of the cumulative hits at about 80,000 s because this process could be highly emissive of the acoustic emission.

The cumulative hits behaved differently in 1000 mg/L and 1200 mg/L of NaCl. In the 1000 mg/L NaCl solution, the cumulative hits increased quickly with time and suddenly vised at about 20,000 s, indicating a breakdown of the passive film in a localized area. Then, the cumulative hits increased slowly with time. However, in 1200 mg/L NaCl, the cumulative hits increased rapidly at the beginning

then gradually and slowly with time, indicating a relatively stable development of the corrosion attack on the surface after 5000 s. Before that time, the pitting corrosion started from the first second and propagated very fast. On the other hand, the OCP stabilized (even a slightly increasing trend) at a certain time in 1000 mg/L (65,000 s) and 1200 mg/L (30,000 s), which indicated the stage of the stable propagation. The high emissive period of AE was before this stage in these two solutions. The results of the AE monitoring and OCP monitoring were in agreement. It was noticed that the significant increase in the AE hits was earlier than the sharp drop of the OCP in these two experiments. It revealed that the AE monitoring is more sensitive to the corrosion evolution compared with the OCP monitoring.

It is interesting that the cumulative hits kept increasing after the stabilization of the OCP curve. The nature of the pitting corrosion is randomly sporadic and stochastic [34]. Therefore, the initiation, accelerated the propagation, and the stable growth/ re-passivation (the corrosion in the pit stopped) of the pitting corrosion is hard to predict. In most circumstances, pitting corrosion happens and propagates at some localized sites independently. The OCP behavior represents the general thermodynamic property of the whole interface, which depends on whether the pitting is meatastable or stable, in which stage—initiation, accelerated propagation, stable growth, or re-passivation—and even the number of active pits. While the increase of cumulative hits stands for the new corrosion events at any local area. These corrosion events refer to any event during corrosion initiation, propagation, and stable growth. While for the whole interface, such corrosion events may have no apparent influence on the OCP in the stable growth stage. However, only the AE signal amplitude of these events exceeds the threshold of the AE acquisition setup, and the recording of AE data can be triggered. That is, the cumulative AE hits in Figure 2 only increased for such corrosion events that can generate an AE signal with an amplitude equal or higher than 27 db.

Figure 2b shows the cumulative absolute energy changes over the time of the AE monitoring. Basically, the higher the AE hits' activity, the faster the increase of the absolute energy than the other stages in the experiments, as shown in Figure 2b, except in the 1200 mg/L NaCl solution. Not only did the acoustic emission of the corrosion event depend on the corrosion stages, and also on the corrosion morphology and other factors, such as the propagation route of pits (i.e., go through the crystalline grain or boundary), but so did the energy of the AE. Although energetic AE signals in pitting normally correspond to the propagation [23,24] and the bursting of the hydrogen bubble, the formation and peeling of thick corrosion products may also generate energetic AE signals if they exist in the 1200 mg/L NaCl solution.

As indicated from the above discussion, the AE events are directly associated to the corrosion phenomena, such as passive film breakdown, hydrogen evolution, pit propagation, etc., compared with OCP monitoring, which only reflects the general electrochemical thermodynamic information. Thus, all the stages of corrosion can be detected by AE monitoring. However, the OCP is only sensitive to the corrosion initiation stage because only the corrosion initiation experiences a significant change in the corrosion potential.

*3.2. Surface Morphology Monitoring of Pitting Corrosion*

Figure 3 shows the surface morphologies from in-situ monitoring of the propagation of pitting corrosion in different $Cl^-$ concentrations at different time intervals of 0 s, 40,000 s, 60,000 s, and 80,000 s, respectively. The surface morphology monitoring experiments were conducted simultaneously with the OCP and AE measurements. After the experiments, the surface morphologies of pits after the removal of loosened corrosion products were observed by SEM, as shown in Figure 4.

In the presence of 500 mg/L NaCl, several pits can be observed on the surface after 40,000 s of immersion. It can be seen from the images that the pits did not grow as the immersion time increased, indicating the pits re-passivated and would not propagate any more. SEM images also proved, as shown in Figure 4a,b, that the pits were open and shallow, without the cover of corrosion products. This type of pitting emitted weak AE signals because it was caused by localized anodic

dissolution, which is not an effective generation source of AE. Additionally, some of them may have been too weak to pass the threshold of 27 dB in the acquisition setup.

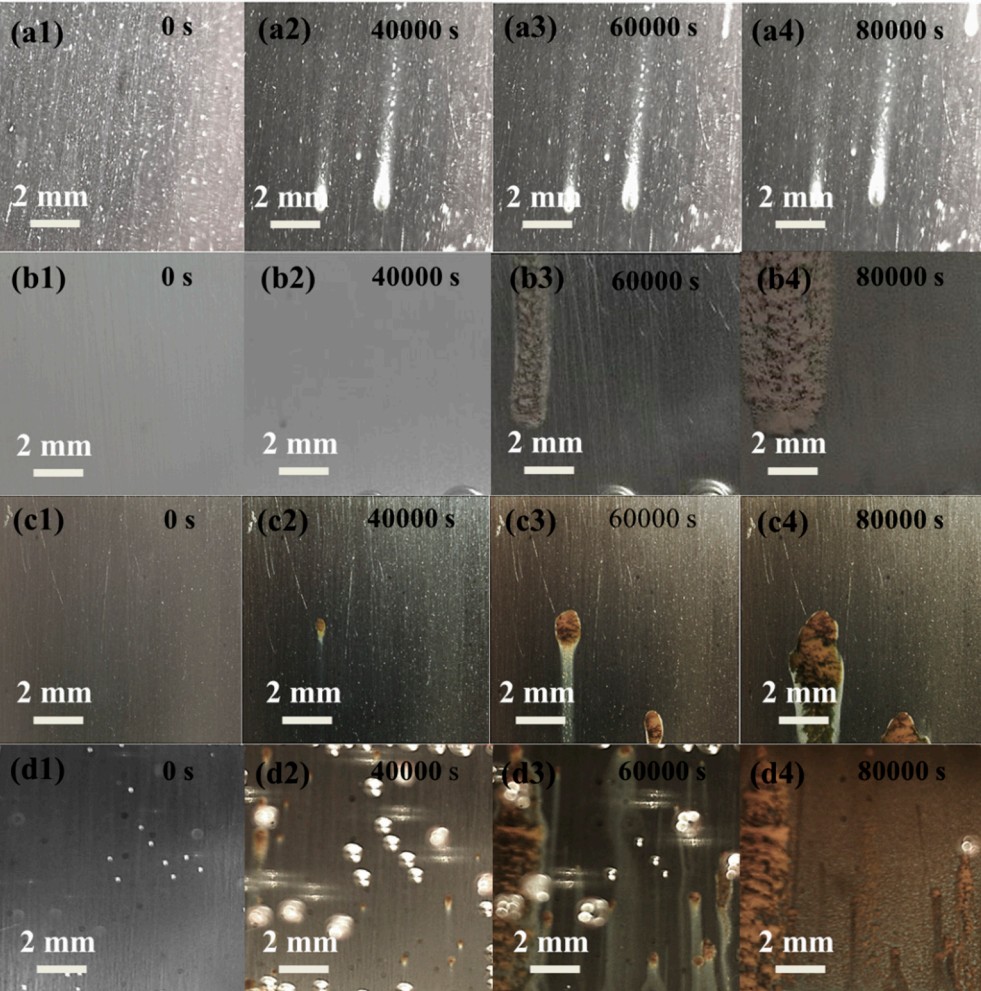

**Figure 3.** Surface morphologies from in-situ monitoring of the propagation of pitting corrosion of carbon steel with an optical microscope in the solutions with different concentrations of chloride ions: (**a1**), (**a2**), (**a3**), (**a4**) 500 mg/L NaCl, (**b1**), (**b2**), (**b3**), (**b4**) 800 mg/L NaCl, (**c1**), (**c2**), (**c3**), (**c4**) 1000 mg/L NaCl, (**d1**), (**d2**), (**d3**), (**d4**) 1200 mg/L NaCl.

No apparent pits can be observed on the carbon steel surface by an optical microscope before 40,000 s in the presence of 800 mg/L NaCl. However, the occluded cell was formed on the specimen surface. This was the reason for the increase of cumulative hits at the beginning of the experiment. After 40,000 s, the two pits gradually initiated and propagated on the surface. The pits continued to grow as the immersion time increased and the corrosion damage area at 80,000 s increased. This behavior was in good accordance with the variation of cumulative hits. In addition, many corrosion products can be observed on the damage area. The occluded environment beneath the corrosion products formed, which accelerated the propagation of the pitting corrosion. SEM images showed some corrosion products that were covered on the pit, as shown in Figure 4c,d. The carbon steel beneath the cover continued to dissolve in the occluded area, which corresponded to the continuous increase of the cumulative hits in Figure 2.

The corrosion morphology evolution and SEM images of the specimen in 1000 mg/L of neutral NaCl solution are shown in Figure 3c1–c4 and Figure 4e–f. Two apparent pits can be observed on the specimen surface at 40,000 s, which continued to grow with immersion time. Many corrosion products can be observed on the damaged area at 80,000 s. The SEM images show that there were some

shallow and small pits on the sample surface. However, because the solution became more corrosive, the corrosion started earlier and its rate increased. As a result, higher AE activity was recorded at the beginning of the monitoring and the AE hits over a short duration near 20,000 s showed a high absolute energy.

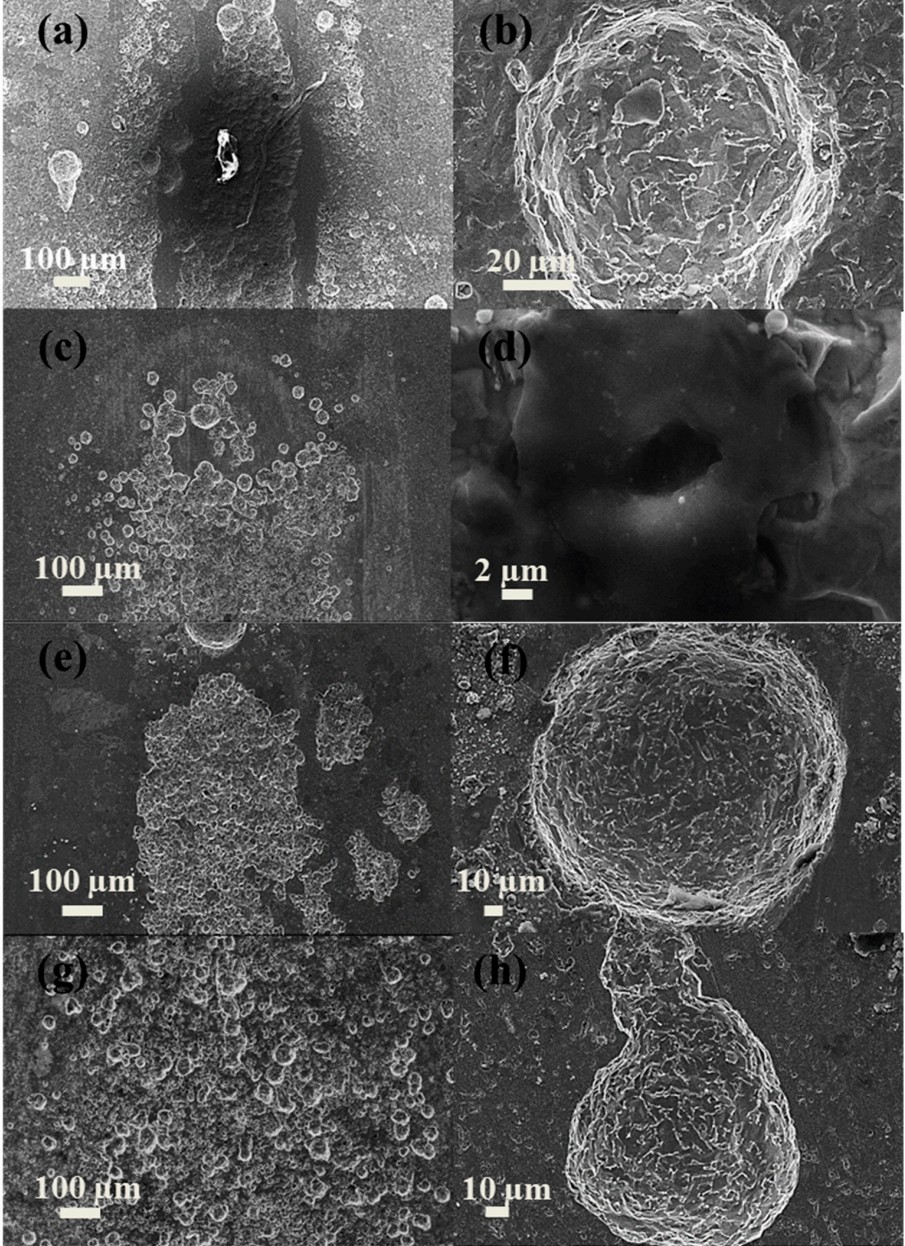

**Figure 4.** SEM morphologies of Q235 carbon steel after immersion for 86,400 s in 2000 mg/L NaHCO$_3$ and NaCl concentration of: (**a**) and (**b**) 500 mg/L NaCl, (**c**) and (**d**) 800 mg/L NaCl, (**e**) and (**f**) 1000 mg/L NaCl, (**g**) and (**h**) 1200 mg/L NaCl, loosened corrosion products were removed.

As implied by the OCP, the sample of Q235 carbon steel was corroded as soon as it was put in the 1200 mg/L NaCl solution, as shown in Figure 3d1–d4. Combined with the cumulative hits curve in Figure 2, the corrosion initiation and propagation were very fast from the first second. Correspondingly, the highest AE activities were in the beginning period, but these did not have a high energy. This can be attributed to the shallow and open shapes of the pits, as shown in Figure 4g–h. The corrosion mechanism of the carbon steel is due to the general corrosion at high Cl$^-$ concentrations while pitting corrosion occurs at low Cl$^-$ concentrations. Thus, in the NaHCO$_3$ solution with 1200 mg/L NaCl,

the localized corrosion tendency was weakened. Finally, most of the surface of the test sample was covered by corrosion products, as shown in Figure 3d4. Nevertheless, the non-uniform coverage of the corrosion products on the carbon steel surface enhanced the occluded effect, which could promote localized corrosion and AE generation.

### 3.3. Waveform Processing

The acoustic signals from experiments were recorded by the AE system. Figure 5a shows a typical waveform of the acoustic signal of carbon steel in NaHCO$_3$ + NaCl solution. It shows that some information of the waveform, the beginning and end, was not useful. So, additional processing of the signals was necessary for the preparation of further analysis. Herein, three steps, including pre-trigger removal, tail-cutting, and shape preserving interpolation (SPI), were applied to process the waveform in Matlab. In pre-trigger removal, a value tells the software how long to record (in μsec) before the trigger point (the point at which the threshold is exceeded), and is aimed at removing digital noise from the acquisition. Tail-cutting was used to remove the "zero-padding", which may have been applied at the end of some waveforms during the acquisition process (Figure 5b). SPI was an effective tool to ensure that each waveform had the same number of points to be stored, which is based on the Piecewise Cubic Hermite Interpolating Polynomial (PCHIP) technique and the Weierstrass approximation theorem. The waveform after processing is shown in Figure 6. In this study, wavelet denoising was performed by the Matlab wavelet tool box (Figure 7). The treated waveform by denoising was stored for the processing and extraction of the parameters.

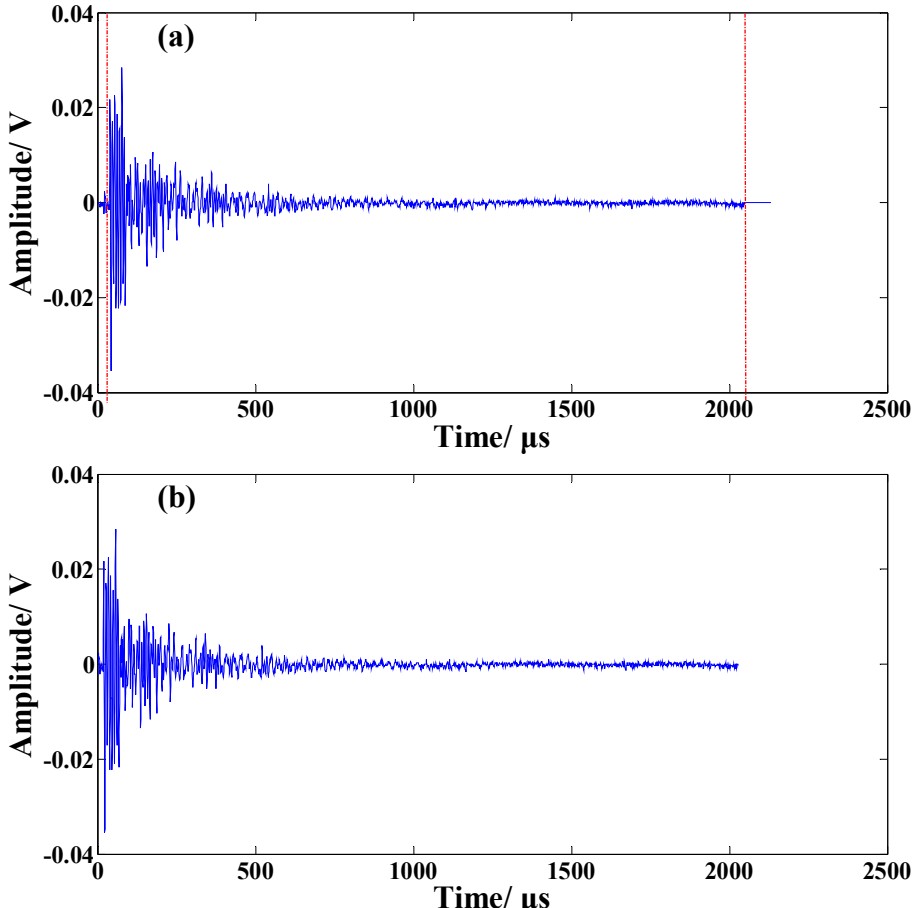

**Figure 5.** The waveform processing of a typical acoustic signal obtained from Q235 carbon steel in NaHCO$_3$ + NaCl solution: (**a**) original, (**b**) pre-trigger removal and tail cutting.

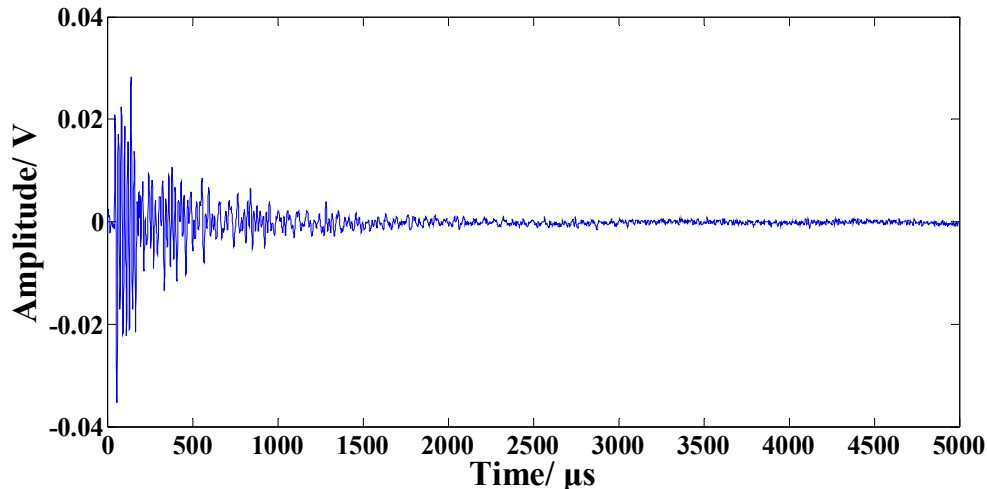

**Figure 6.** The processed waveform of a typical acoustic signal obtained from Q235 carbon steel in NaHCO$_3$ + NaCl solution (after SPI and wavelet denoising).

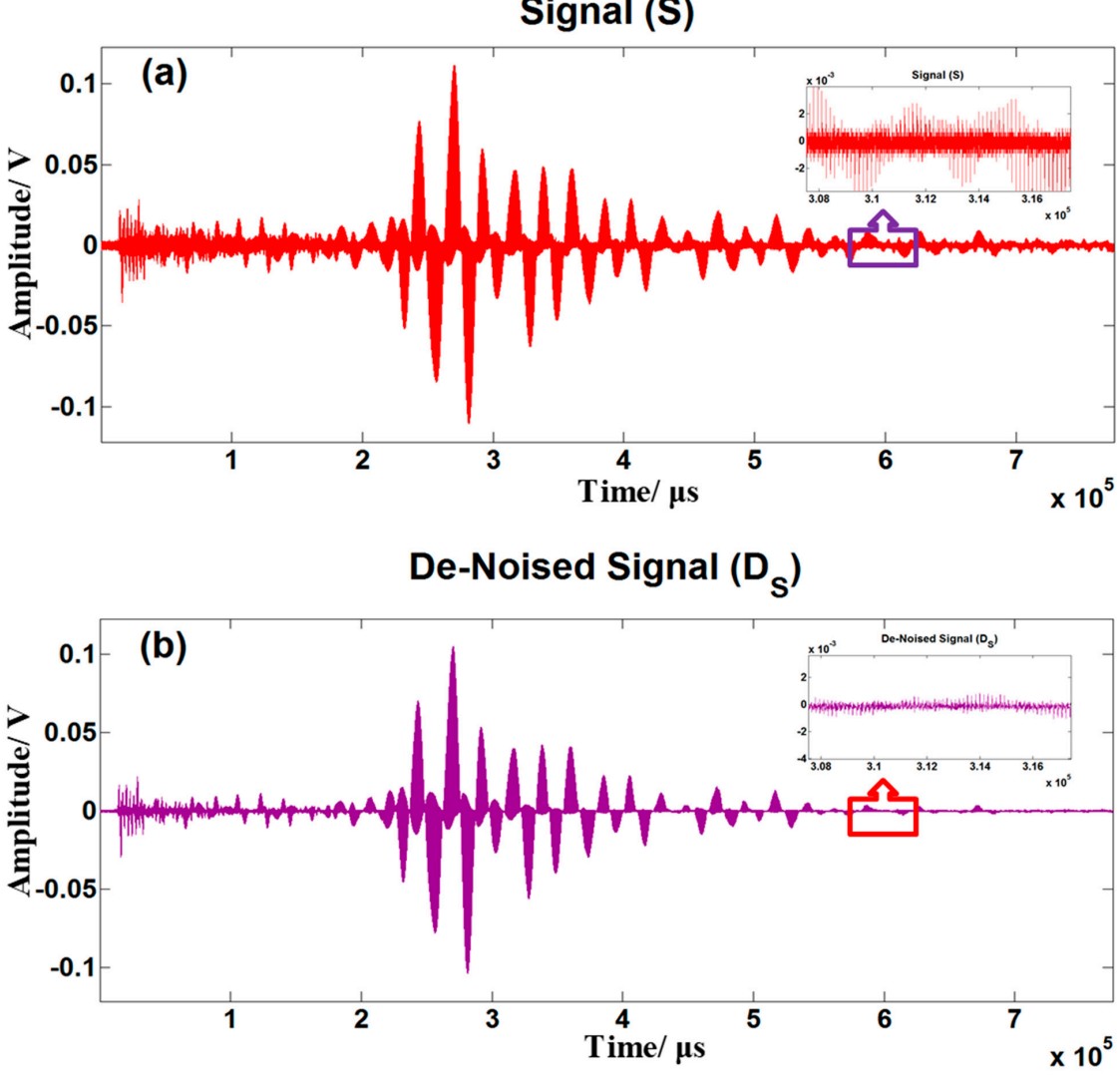

**Figure 7.** Denoising method for a typical signal of the acoustic emission of carbon steel in NaHCO$_3$ + NaCl solution. (**a**) Signal before denoising, (**b**) signal after denoising.

### 3.4. Clustering Analysis

A function was established to input waveform information, time and system information including the maximum duration, hit definition time (HDT), hit lockout time (HLT), peek definition time (PDT), sampling frequency, and the threshold, to Matlab. Some AE parameters were used for the clustering analysis, for instance, the "count", the "duration", and the "average frequency". The count of the AE event is the number of times the signal rises and crosses the threshold of the AE setup. The duration of the AE event is the time between the rising edge of the first count and the falling edge of the last count. The average frequency (Hz) was calculated from $N_{AE}/t_{AE}$ to estimate the characteristic frequency of an AE event. These parameters were recorded by the AE acquisition system in-situ and extracted in Matlab after waveform processing, respectively.

After the extracting feature, these new parameters were plotted in additional correlation diagrams. The comparison between the original parameters and the new parameters under different corrosion conditions are shown in Figures 8 and 9. These results showed that the performing of the developed signal processing can effectively classify AE signals in the correlation diagram.

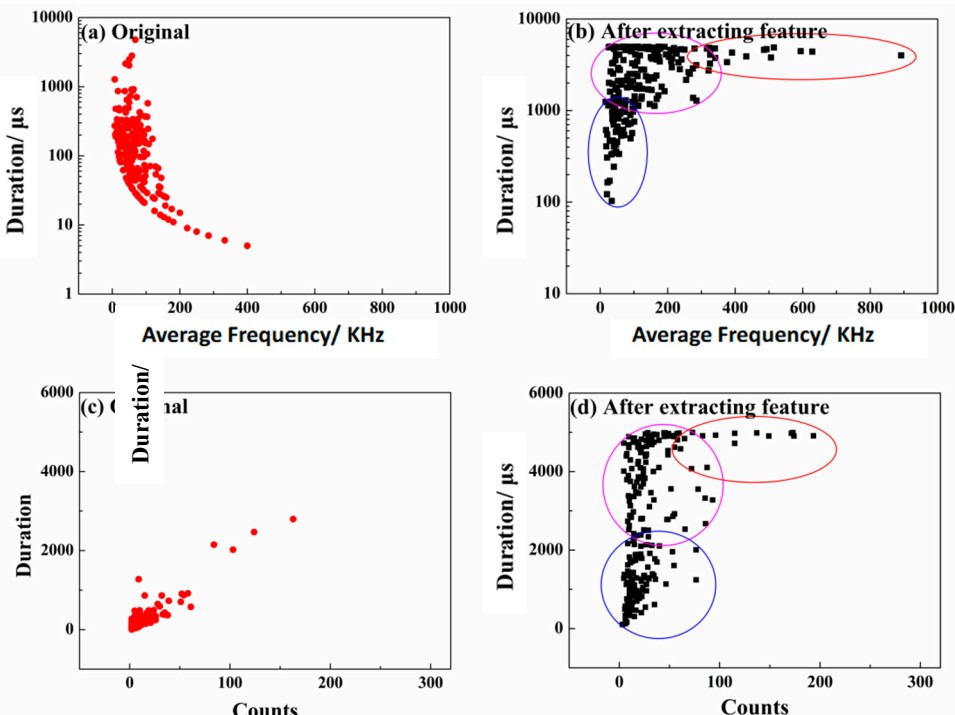

**Figure 8.** Analysis of the parameters of the signals of Q235 carbon steel in 2000 mg/L NaHCO$_3$ + 800 mg/L NaCl: (**a**) and (**c**) original relationship of the duration-average frequency and duration-counts, (**b**) and (**d**) relationship of the duration-average frequency and duration-counts after the extracting feature.

Based on the analysis of the relationship between the duration and average frequency of the AE waveforms, the AE signals of Q235 carbon steel in different corrosive media (2000 mg/L NaHCO$_3$ + 800 mg/L NaCl and 2000 mg/L NaHCO$_3$ + 1000 mg/L NaCl solution) during the pitting process were classified into three types: High duration with high frequency, high duration with low frequency, and low duration with low frequency. In addition, the three groups of signals were identified according to the duration versus counts correlation diagram: Low duration with low counts, high duration with low counts, and high duration with high counts.

The representative waveforms of the three types of signal are shown in Figure 10. The low duration (<2000 μs) with low counts (<50) signals and high duration (>2000 μs) with low counts (<50) signals appeared most frequently at the beginning of the experiment, indicating that they belong

to the corrosion event of the surface breakdown. They were the burst signal (Figure 10a, duration 492 μs, counts 23) and asymmetric signal (Figure 10b, duration 2732 μs, counts 44). Most of the high duration (>2000 μs) and high counts (50–100) signals occurred during the accelerated propagation of pitting corrosion. They were resonant signals (Figure 10c, duration 2336 μs, counts 98). In the stable growth process of the pitting corrosion after the accelerated propagation, the AE waveforms were also resonant signals. However, the signal feature was more complex than that in the accelerated propagation process (Figure 10d, duration 1878 μs, counts 77), which may be due to the combination of the breakdown of corrosion products and the growth of the pits at the same time. The above result is consistent with other studies on the acoustic emissions of pitting corrosion [23,35,36].

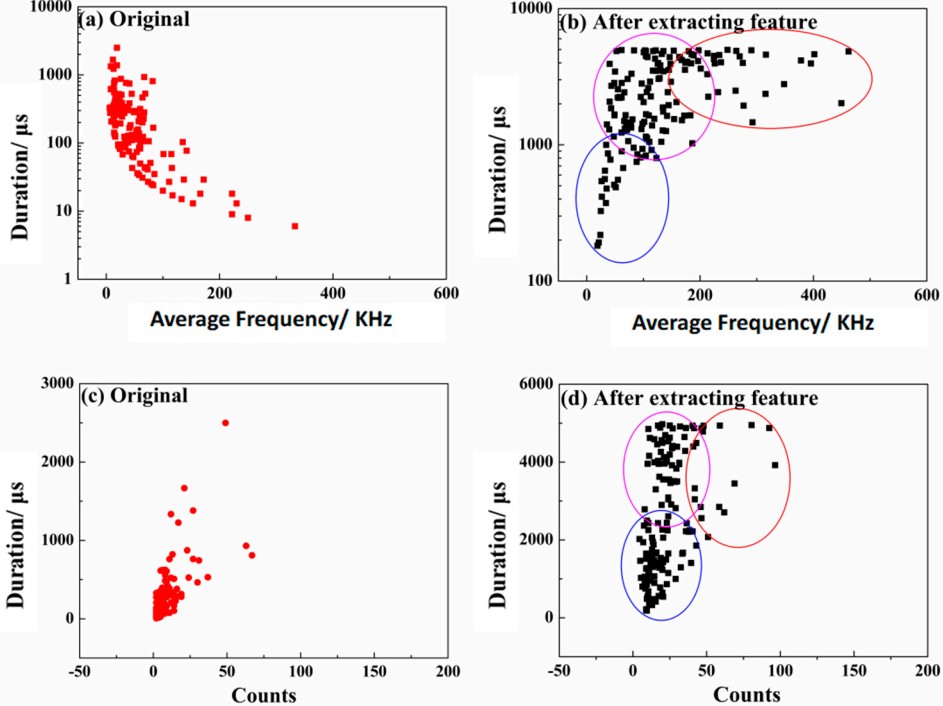

**Figure 9.** Analysis of the parameters of the signals of Q235 carbon steel in 2000 mg/L NaHCO$_3$ + 1000 mg/L NaCl: (**a**) and (**c**) original relationship of the duration-average frequency and duration-counts, (**b**) and (**d**) the relationship of duration-average frequency and duration-counts after the extracting feature.

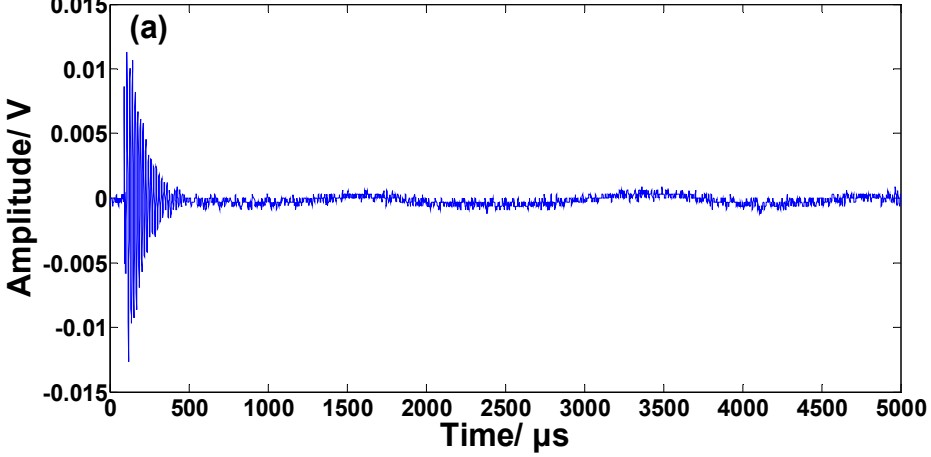

**Figure 10.** *Cont.*

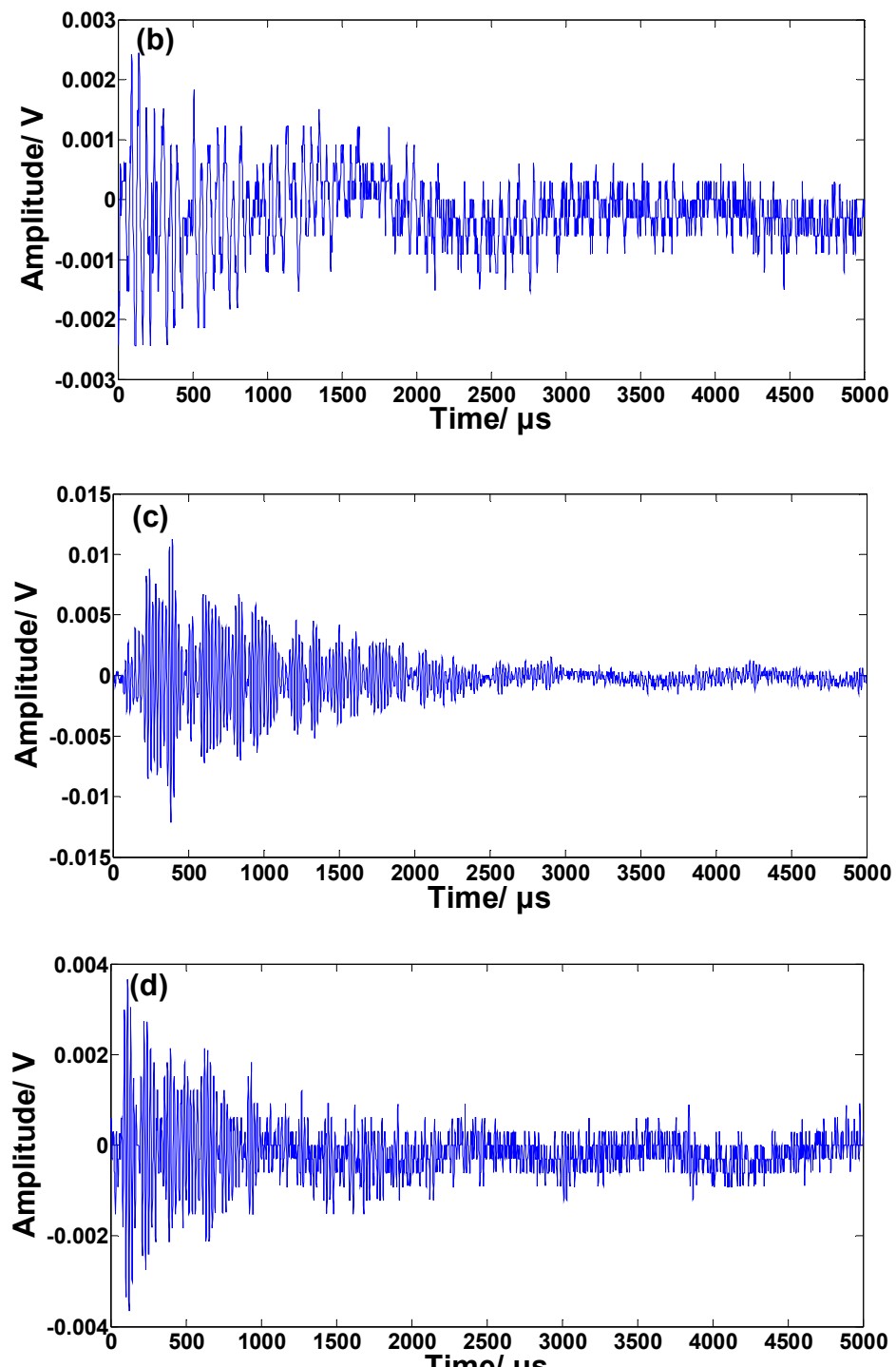

**Figure 10.** Waveforms from top to bottom: (**a**) and (**b**) low duration with low frequency and low counts, (**c**) high duration with low frequency and low counts, (**d**) high duration with high frequency and high counts.

Many researchers have tried to quantitatively correlate AE parameters to corrosion based on parameter recognition or clusters. A good correlation between the cumulative hits of surficial oxide film rupture and the corrosion current density in carbon corrosion was reported [35]. However, to the emission of AE is not only affected by corrosion phenomena, but also by the mechanical condition of steel structure. It is very difficult to determine the corrosion rate by AE in practical settings. In addition, though the recognition and identification of AE signals could offer useful information regarding the

corrosion mechanism, identification of the group of signals that represents the corrosion rate requires further investigation.

*3.5. Corrosion Type Identification*

The typical methods for distinguishing corrosion types includes AE parameters cluster (i.e., k-means) and the ensemble learning method (i.e., random forest). These methods only utilize some parameters of the AE waveform (raw or pre-treated) or rely on big data. To identify the different corrosion types in the corrosion process fast and effectively, a 2D pattern recognition algorithm was established by Matlab. This processing algorithm provides a new feasible method for the identification of different acoustic emission data directly from the waveform of the AE signal without the extraction of AE parameters.

In 2D pattern recognition analysis, the luminance information of the object surface is related to the illumination and reflection coefficient. Otherwise, the structure and illumination of the objects in the scene are independent. The reflection coefficient and the related objects are the same. We can explore the structure information in an image by separating the illumination effects of objects. Here, the object and structure relate to the brightness and contrast as the structure information of the image definition. Because the brightness and contrast of a scene are always changing, we can perform local processing to obtain more accurate results.

The measurement of similarity is composed of three modules: Luminance, contrast ratio, and structure. Their functions are defined as follows.

First, for the discrete signal, we used the average gray level as a measure of the intensity of the estimation, as function (1) shows, and the brightness contrast function, $L(x, y)$, is a function about $\mu_x$ and $\mu_y$:

$$\mu_x = (1/N) \sum_1^N x_i \tag{1}$$

Then, the average gray value should be removed from the signal based on the measurement system. For the discrete signal, $x - \mu_x$, the standard deviation can be used to do the contrast measure, such as function (2) shows, and the contrast function, $c(x, y)$, of the contrast is a function of $\sigma_x$ and $\sigma_y$:

$$= \left( 1/N - 1 \sum_{i=1}^N (x_i - \mu_x)^2 \right)^{1/2} \tag{2}$$

Next, the signal is divided by its standard deviation, and the contrast function structure is defined as a function of $\frac{(x - \mu_x)}{\sigma_x}$ and $\frac{(y - \mu_y)}{\sigma_y}$.

Finally, the three contrast modules are combined into a complete similarity measure function:

$$S(x, y) = (l(x, y), c(x, y), s(x, y)) \tag{3}$$

$S(x, y)$ should meet the following three conditions:
Symmetry: $S(x, y) = S(y, x)$;
bounded property: $S(x, y) \leq 1$; and
maximum uniqueness: $S(x, y) = 1$, when and only when $x = y$.

Then, the three contrast functions are defined, and finally they are combined to obtain the following similarity function:

$$SSIM(x, y) = [l(x, y)]^\alpha [c(x, y)]^\beta [s(x, y)]^\gamma \tag{4}$$

When $\alpha = \beta = \gamma = 1$, a function would be obtained as follows:

$$SSIM(x, y) = (2\mu_x\mu_y + C_1)(2\sigma_x\sigma_y + C_2) / \left( \mu_x^2 + \mu_y^2 + C_1 \right) \left( \sigma_x^2 + \sigma_y^2 + C_2 \right) \tag{5}$$

Other corrosion types were monitored by AE, aiming to calculate the similarities among them. The pitting corrosion of stainless steel in the presence of different concentrations of chloride ions and the uniform corrosion of carbon steel in sulphuric acid solution was monitored by the same device as Figure 1. The specimens of the crevice corrosion of stainless steel were pretreated based on ASTM G39-1999(2011) and monitored by the method of Kim [18], and the AE waveforms of different types of corrosion were arranged in a time sequence. Figure 11 shows the continuous AE waveform graphics of the different corrosion experiments. These continuous AE waveforms were obtained by arranging the waveform of each AE event one by one in the time sequence. These waveform graphics were imported into Matlab at first, then the 2D pattern recognition algorithm was performed as described by formula (1) to (5) to obtain the similarities of the different graphics. The calculation results are exhibited in Table 3.

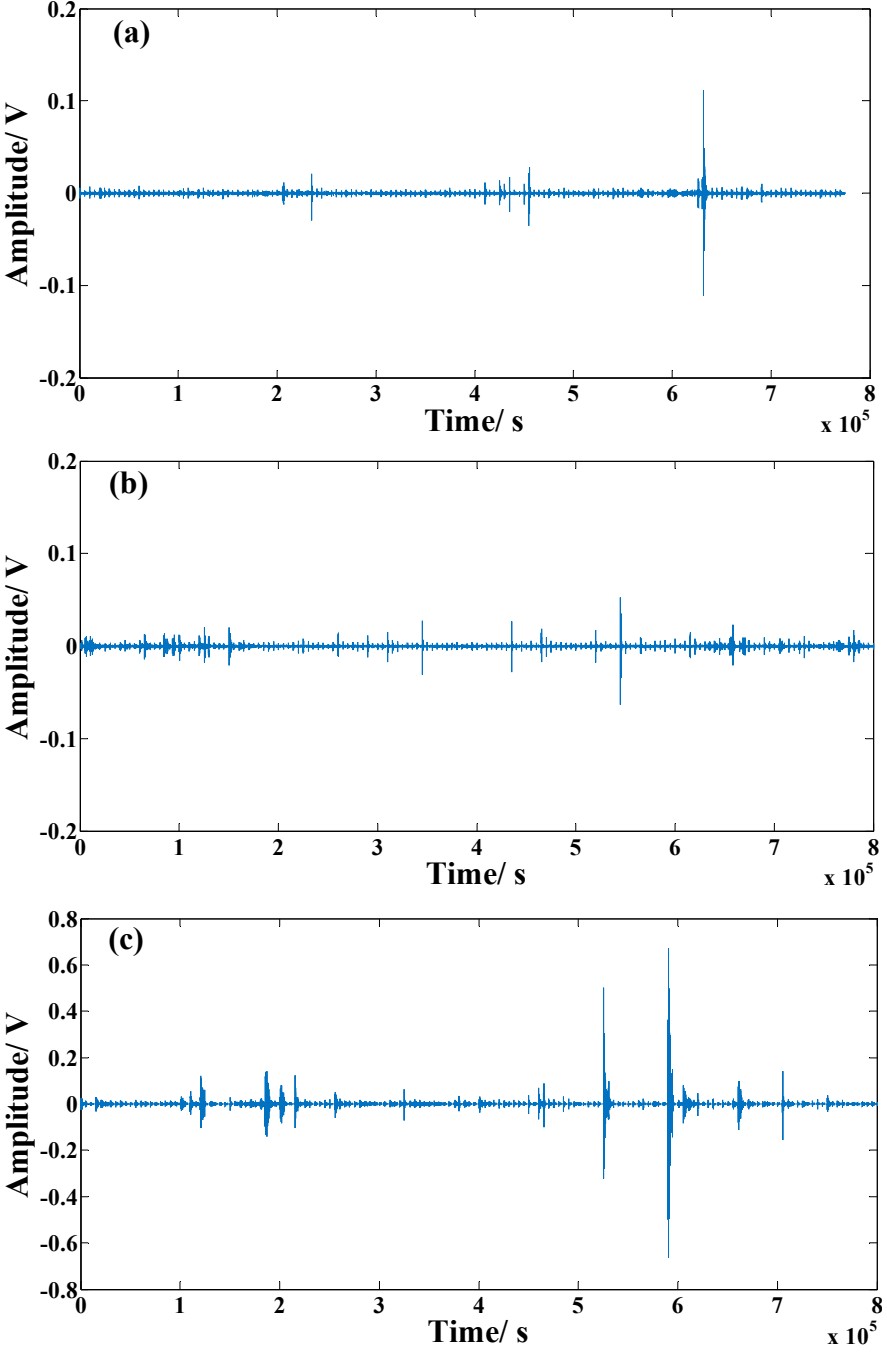

**Figure 11.** *Cont.*

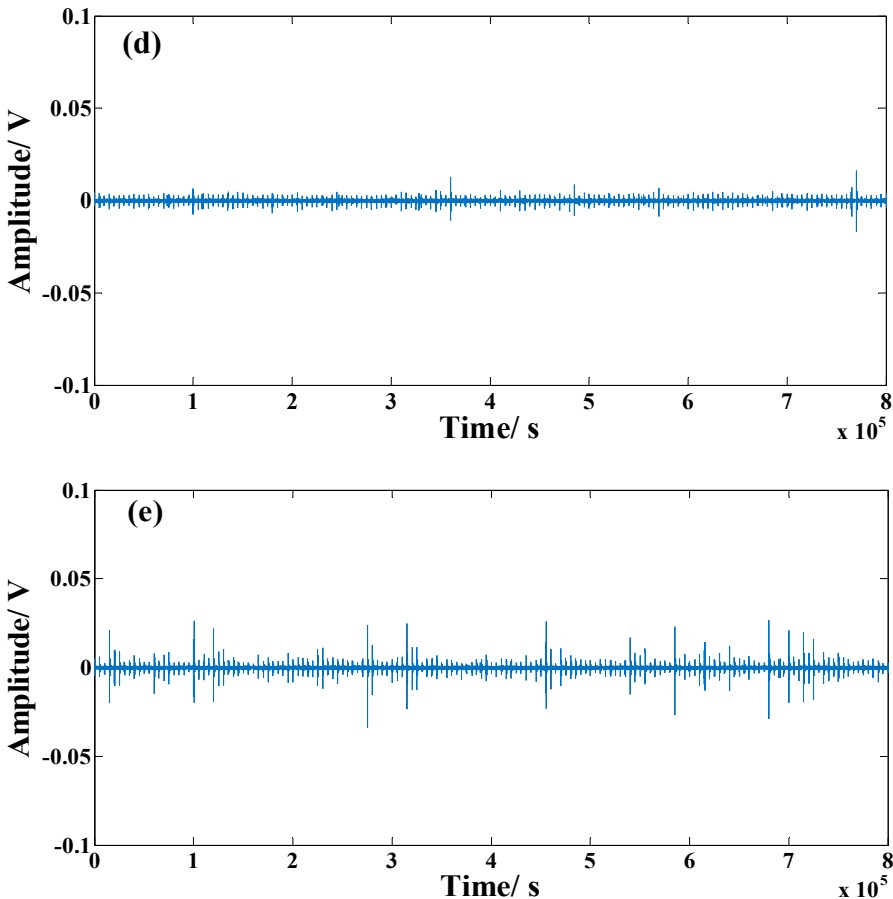

**Figure 11.** Different continuous AE graphics: (**a**) pitting corrosion of carbon steel, (**b**) pitting corrosion of stainless steel, (**c**) crevice corrosion of stainless steel, (**d**) uniform corrosion of carbon steel, (**e**) noise.

**Table 3.** The similarity of the different types of corrosion.

| | Pitting Corrosion Carbon Steel | Crevice Corrosion Stainless Steel | Uniform Corrosion Carbon Steel | Noise |
|---|---|---|---|---|
| Pitting corrosion Stainless steel | 83.23% | 78.03% | 50.12% | 27.53% |
| Pitting corrosion Carbon steel | | 85.51% | 52.16% | 48.74% |
| Crevice corrosion Stainless steel | | | 35.41% | 45.13% |
| Uniform corrosion Carbon steel | | | | 42.13% |

According to the calculation results, there was a high similarity between the pitting corrosion of carbon steel and that of stainless steel. The occluded cell of the pitting corrosion on carbon steel was

not stable in the performed experiments. The repeating process of the occluded cell formation and rupture caused the open pits on the surface of the carbon steel. In contrast, the pitting corrosion of stainless steel had a relatively strong occlusion effect, such that the corrosion pit of the stainless steel often had a small mouth with relatively larger depth. However, these two kinds of pitting corrosion were all caused by the occlusion effect, from which they had a similar acoustic emission process (83.23% similarity) as shown in Table 3. In addition, the corrosion process of the 2D images of the pitting corrosion and crevice corrosion also had a high degree of similarity (73.03%), which is related to their corrosion mechanisms being relatively similar [34]. On the other hand, the graphics of the uniform corrosion had a low similarity with the other types of corrosion, indicating that this data processing approach can identify localized corrosion from uniform corrosion. In addition, the AE graphic of the artificial noise from the stirring of the solution also had low similarities with all the corrosion graphics.

## 4. Conclusions

The pitting corrosion of Q235 carbon steel in $NaHCO_3$ + NaCl solutions was studied by in-situ monitoring of the AE technique and OCP simultaneously. The concentration of NaCl had a pronounced influence on the OCP evolution. In 500 mg/L NaCl, the OCP varied in a very narrow range, indicating a relatively stable state. However, with the increase of the NaCl concentration, the OCP dropped significantly and stabilized at a rather negative potential.

The AE monitoring results were in accordance with the results of the OCP monitoring. However, the OCP only presented thermodynamic information of the corrosion in the interface, while the AE sensor detected the breakdown of the passive film and small damage in the occluded pits under the metallic surface, and the relative events.

The AE signals of pitting corrosion on carbon steel were classified into three types by waveform parameters clustering after AE waveform processing, including pre-treatment, shape preserving interpolation, and denoising. The result indicated that the developed signal processing is a highly efficient method for the classification of AE signals and preparation of the waveform for further data analysis.

A method based on 2D pattern recognition was established for the identification of different types of corrosion in Matlab. The analysis results showed that the method can be used to distinguish between uniform corrosion and localized corrosion effectively, while it was not very effective for distinguishing different localized corrosion.

The AE technique could be applied to in-situ monitoring of the corrosion of carbon steel or stainless steel containers in different industries and for the reinforcement of steel bars for construction. However, the main obstacle to the use of AE monitoring is environmental noise, which make data processing difficult.

**Author Contributions:** Conceptualization, J.T. and H.W.; methodology, J.T. and H.W.; software, J.T. and H.W.; investigation, J.T., H.W. and G.C.; data curation, J.L.; writing—original draft preparation, J.T. and Y.W.; writing—review and editing, J.T. and Y.W.; visualization, J.T. and J.L.; supervision, H.W.

**Funding:** This research was funded by Applied Basic Research Programs of Science and Technology Department of Sichuan Province, grant number 2017JY0044 and Project Funding to Scientific Research Innovation Team of Universities Affiliated to Sichuan Province, grant number 18TD0012.

**Conflicts of Interest:** The authors declare no conflict of interest.

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
