# Peer review of "In-Situ Monitoring and Analysis of the Pitting Corrosion of Carbon Steel by Acoustic Emission"

_applsci, doi:10.3390/app9040706_

Round 1
Reviewer 1 Report
Since this paper refers existing researches on the correlation between the AE signal and the corrosion phenomenon, the novelty of this research is considered to be your analysis method of the AE signal.
However, the description in "3.4 Clustering analysis" is insufficient, and correction is necessary.
What are the durations, counts, average frequency, which are labels of axes in Fig. 8 and 9? There are no units in these labels.
If the unit of average frequency is kHz, considering that the frequency bandwidth of the analog filter is 100-400 kHz, there are many plots outside this bandwidth.
Furthermore, there are no quantitative criteria describing to classify the group of plots into three groups.
The subjective classification of authors such as low and high is applied also in Fig. 10, and it is impossible for the reader to verify these results.
The logical development in "3.5 Corrosion type identification" is also unclear, and correction is necessary.
Did you compare the type of corrosion distinguished by AE signal with the teacher data that was pattern-recognized beforehand and investigated those similarity?
Even if there are existing researches, criteria for distinguishing corrosion types with AE signals should be described to some extent.
It is strongly recommended to clearly show the specific flow for obtaining the results in Table 3.
Author Response
Thanks very much for the valuable comments from the reviewer. The improvement of the manuscript has been made according to the comments and questions. The responses to each comment or question are in the attachment.

Reviewer 2 Report
The work is very interesting and very well presented from a scientific point of view. Inevitably the treatment of acoustic emission signals is very complex for interested readers and experts in steel corrosion, and not in AE. It would therefore be desirable to have a more detailed description of this process. In any case, in my opinion the authors present a well-founded demonstration that their method of acoustic emission is capable of detecting and corrosion of carbon steel in that medium and even more of distinguishing between a uniform corrosion process or localized corrosion. However, from the point of view of a real application of this methodology to the control of corrosion of carbon steel structures, the important parameter is not so much the OCP as the corrosion current density. Parameter that effectively determines the loss of material. In my opinion, the biggest flaw in the manuscript that the authors should explain is whether in their opinion it is possible to correlate the AE with the corrosion rate of carbon steel.
Author Response

(The authors gave the same response as above.)

Reviewer 3 Report
The work is too interesting and well oragnized. This work can be accepted after minor revisions. The following points have to be considered into this manuscript before acceptance:
What about passivation and re-passivation behavior/role of such materials investigated.
Could the polarization curves (Evan Diagrams) be included in this work for more clearness.
How this technique could be applied practically and for which applications (main application).
What is the main obstacle of using this technique (acoustic emission)? (probability of Passive layer break down??.
The main difference between AE technique and OCP one. The most attarctive points should be clear in this work.
Author Response

(The authors gave the same response as above.)

Round 2
Reviewer 1 Report
Thank you for your response. I understood the difficulty of classification of corrosion and your analysis procedure.
Your paper seems generally good. Please consider the following matters.
1. I recommend that you add the content of your answer on durations, counts, average frequency to your paper as well.
2. The unit of average frequency in Fig.8 and Fig.9 seems to be kHz instead of Hz.
Plots of 50 kHz or less and 400 kHz or more are seen in these figures. Please let me check again if your system with frequency bandwidth from 100 to 300 kHz can measure these data.
3. The criteria of corrosion classification is the important know-how of the authors. I expect to establish quantitative judgment criteria in your future work.
4. I think that the differences between Fig. 10 (a)-(d) should be described more quantitatively.
Author Response
Dear reviewer,
Thanks a lot for the valuable comments and suggestions. The improvement of the manuscript has been made according to them.
Comments and Suggestions for Authors
Thank you for your response. I understood the difficulty of classification of corrosion and your analysis procedure.
Your paper seems generally good. Please consider the following matters.
1. I recommend that you add the content of your answer on durations, counts, average frequency to your paper as well.
A: Thanks for the suggestion, we have added the relative content in the revised manuscript (line 305-311). These parameters were recorded by AE acquisition system in-situ and extracted in Matlab after waveform processing, respectively.
2. The unit of average frequency in Fig.8 and Fig.9 seems to be kHz instead of Hz.
Plots of 50 kHz or less and 400 kHz or more are seen in these figures. Please let me check again if your system with frequency bandwidth from 100 to 300 kHz can measure these data.
A: Very sorry for our carelessness. The unit should be kHz and they were corrected in the revised paper. For the bandwidth of 100 to 300 kHz, which is the filter applied for the acoustic signal attenuation in other frequency domain, but it can not make all peaks in other frequency domain disappearing. The features of waveform highly depend on the nature of acoustic emission event.
3. The criteria of corrosion classification is the important know-how of the authors. I expect to establish quantitative judgment criteria in your future work.
A: Thanks very much for your encouragement! We are working hardly for defining the quantitative judgment criteria in AE monitoring. In the previous works, for instance “Kim, Y.P.; Fregonese, M.; Mazille, H.; Féron, D.; Santarini, G. Ability of acoustic emission technique for detection and monitoring of crevice corrosion on 304L austenitic stainless steel. NDT E Int. 2003, 36, 553–562” reported that the corrosion area value was in accordance with the cumulative AE events. However, when I worked with Pr. Fregonese in MATEIS to repeat the same experiment, we found former conclusion was not reliable. Then I found the stress level will obvious affect the acoustic emission activity of crevice corrosion (J,L. Tang, Eurocorr2013, Lisbon).For pitting, both the initiation and the propagation of the pit can generate the AE. The propagation of pit is difficult to measure and its acoustic emission also depends on the stress level theoretically. Recently, we designed an experiment illustrated as below. Different stress can be applied to steel during corrosion. We think the present work can help us to make progress in quantitative judgment criteria.
4. I think that the differences between Fig. 10 (a)-(d) should be described more quantitatively.
A: Thanks very much for the suggestion. The relative text (line 332-340) was revised according to the suggestion. The parameter values were added in the revised paper. The criteria for duration and counts classification were defined quantitatively.
We tried our best to improve the manuscript and made some changes of this manuscript according to the comments and suggestions from the reviewer. And these changes were marked in blue in the revised manuscript.
We appreciate for Editors and Reviewers’ warm work earnestly, and hope that the correction will meet approval.
Yours sincerely,
Junlei Tang
